# Maternal deaths before and during COVID-19 pandemic: Causes and avoidable factors in a tertiary hospital in South Africa, 2018–2022

Ongombe Lunda[1]*, Lawrence Chauke[1], Ghada Daef[1,2], Gbenga Olorunfemi[3], Nnabuike Chibuoke Ngene[1,4]

1 Department of Obstetrics and Gynaecology, School of Clinical Medicine, Faculty of Health Sciences, University of the Witwatersrand, Johannesburg, South Africa, 2 Department of Obstetrics and Gynaecology, Klerksdorp Hospital, Klerksdorp, North West, South Africa, 3 Division of Epidemiology and Biostatistics, School of Public Health, University of the Witwatersrand, Parktown, Johannesburg, South Africa, 4 Department of Obstetrics and Gynaecology, University of the Witwatersrand and Rahima Moosa Mother and Child Hospital, Gauteng Province, South Africa

* alphongombe@yahoo.fr

## Abstract

### Aim

To determine the institutional maternal mortality ratio (iMMR) and avoidable factors (AVFs) before and during the COVID-19 pandemic in a tertiary hospital in South Africa.

### Methods

This was a retrospective cross-sectional study. We reviewed medical records to compare iMMR and associated AVFs two years before (March 2018 – February 2020) and two years during (March 2020 – February 2022) COVID-19 pandemic.

### Results

Fifty-eight maternal deaths were recorded but available data was 57 (35 before and 22 during COVID-19 pandemic). The highest iMMR per 100,000 live births over a 12-month period was 329.1 before and 201.8 during COVID-19 pandemic. The mean ages were 31.0±6.9 and 31±6.2 years, p=0.822 before and during COVID-19 pandemic, respectively. During COVID-19 pandemic, 40.9% (9/22) were diagnosed with COVID-19. Before and during the pandemic, 71.4% (25/35) and 68.2% (15/22) p=1.0 were admitted into an intensive care unit (ICU), respectively, with corresponding 74.3% (26/35) and 50% (11/22), p=0.026 requiring mechanical ventilation. There was a significant difference in the primary causes of death (p=0.009) for the two periods, with preeclampsia with severe features (22.9%, 8/35) being the leading cause before COVID-19 pandemic compared to COVID-19 (31.8%, 7/22) during the pandemic.

**Data availability statement:** The data cannot be shared publicly because it contains sensitive patients' information. The data underlying the results presented in the study are available via the following institutional contact after their permission: University of the Witwatersrand Human Research Ethics Committee, Faculty of Health Sciences, Philip Tobias Building, Offices 301 – 304, 3rd Floor, Corner York Road and 29 Princess of Wales Terrace, Parktown, 2193, Gauteng Province, South Africa. Phone: +27 11 717 2700. Email: HREC-Medical.ResearchOffice@wits.ac.za. Access to the data does not involve a third-party organization.

**Funding:** The author(s) received no specific funding for this work.

**Competing interests:** The authors have declared that no competing interests exist.

AVFs related to healthcare professionals were the most common occurring in 22/35 and 12/22 of the deceased before and during COVID-19 pandemic, respectively. Before COVID-19 pandemic, the most frequent patient-, healthcare professional-, and administrative-related AVFs were failure to book for antenatal care, administration of wrong treatment, and lack of ICU/high care bed spaces, respectively. During COVID-19 pandemic, the most common patient-, healthcare professional- and administrative-related AVFs were delay in seeking medical treatment, lack of critical care skills, and unavailability of ICU/high-care bed spaces, respectively.

## Conclusion

iMMR was lower during than before the COVID-19 pandemic. AVFs related to healthcare professionals were the most common.

---

## 1. Introduction

The socioeconomic challenges and adverse health outcomes associated with the coronavirus disease 2019 (COVID-19) pandemic have been devastating to many individuals, families, and communities. However, evidence-based interventions such as vaccination and non-pharmacological interventions including regular hand washing, as well as attainment of herd immunity, have largely halted the scourge of the disease. COVID-19 was first reported in China in the city of Wuhan on December 8, 2019 [1]. It is caused by severe acute respiratory syndrome coronavirus-2 (SARS-CoV-2). On March 11, 2020, the World Health Organization (WHO) declared the disease a pandemic [1]. The disease is highly infectious and in Sub-Saharan Africa had a case fatality rate of 3.4% [2]. As of December 23, 2024, it had resulted in 7.1 million deaths worldwide [3]. In South Africa, the first case was reported in the KwaZulu-Natal province on March 6, 2020. The infection rapidly spread to other provinces with the national cumulative confirmed cases per million people increasing to 3 398.46 in six months (as of September 8, 2020) and to 58 068.32 in two years (as of February 6, 2022). By June 6, 2022, the figure had become 63 596.97 per million people and the gradient plateaued [4]. To place this in context, South Africa had a mid-year population of approximately 60 million in 2021 [5]. By June 20, 2022, all the COVID-19 restrictions in South Africa had been lifted [6]. Following a declining trend in the number of deaths and hospitalizations from the disease globally, WHO declared on May 5, 2023 that COVID-19 was no longer a public health emergency of international concern [7]. Some survivors of the disease, however, still struggle to cope with the debilitating effects of long COVID-19. The pandemic disrupted activities of daily living including the usual routines in the health sector and, as a result, healthcare resources had to be redistributed to contain the pandemic [8]. Furthermore, COVID-19 deaths preferentially affected vulnerable groups such as persons with comorbidity (including hypertension and diabetes) and those with immunosuppression [9]. Consequentially, many maternal deaths due to COVID-19 occurred globally including in South Africa [10].

During the pandemic, pregnant women in South Africa with severe COVID-19 were transferred to regional and tertiary hospitals for further management because of the availability of established critical care services at these referral centers. In the context of this referral pattern, the contribution of COVID-19 to maternal deaths in South Africa has not been exhaustively studied. Although data on institutional maternal deaths in South Africa are collated and published every three years in the Saving Mothers Report [10], evidence suggests that the most common causes of and trends in maternal deaths at the national level (based on the consolidated data from various healthcare facilities) may be different from a hospital-specific data review [11]. The 2017 – 2019 Saving Mothers Report suggests that the national institutional maternal mortality (iMMR) ratio increased during the first wave of the COVID-19 pandemic in South Africa, from 98.8 to 126.1 per 100,000 live births [10]. Whether or not iMMR increased in each maternity unit in South Africa during the COVID-19 pandemic remains uncertain.

The iMMR, causes of maternal deaths, and associated avoidable factors (AVFs) before and during the COVID-19 pandemic have not been studied in Klerksdorp/Tshepong Hospital, the only tertiary hospital in the North West province of South Africa. The information gained from such research could assist with the identification and implementation of interventions to prevent future maternal deaths and prepare the health system for future pandemics. The aim of this study, therefore, was to determine the trends in iMMR, causes, and AVFs associated with maternal deaths in a tertiary hospital in the North West province in South Africa over a four-year period (March 1, 2018 – February 28, 2022) comprising two years before and two years during the COVID-19 pandemic.

## 2. Materials and methods

### 2.1 Study design and period

The study was a cross-sectional design utilizing a retrospective medical record review of all in-hospital maternal deaths that occurred before the COVID-19 pandemic (March 1, 2018 – February 28, 2020), and during it (March 1, 2020 – February 28, 2022). The study covered the 24-month period immediately before COVID-19 was diagnosed in South Africa in March 2020 and the immediate 24-month period during the COVID-19 pandemic. Data extracted/collection from the medical records of the deceased women commenced on June 1, 2022 and ended on December 31, 2022.

### 2.2 Study setting

The study was conducted in the Klerksdorp/Tsphepong hospital, which is made up of two healthcare facilities namely Klerksdorp Hospital in Klerksdorp town and Tshepong Hospital in Jouberton township in Kenneth Kaunda District, North West Province, South Africa. The two public hospitals are tertiary facilities that function as a complex. The Department of Obstetrics and Gynecology is in Klerksdorp Hospital and conducts 480 deliveries per month. It receives referrals from the district and provincial hospitals as well as 16 primary healthcare clinics in the Kenneth Kaunda Health District and other health districts in the North West province. Four of the primary health care clinics function as midwife-led obstetric Units. Of note, the referral patterns to the hospital did not change during the study period, with all patients presenting with respiratory symptoms suspected to be tuberculosis and those diagnosed with COVID-19 being screened for tuberculosis.

In South Africa, the initial three COVID-19 waves (high rate of SARS-CoV-2 transmission) occurred from April 2020 to September 2021 with the timing of the surges being similar across the provinces in South Africa [12, 13]. There was successive higher positivity of diagnostic test, incidence, and mortality of COVID-19 across the three periods with the ancestral strain, Beta (B.1.351) and Delta (B.1.617.2) variants being responsible for the first, second, and third waves, respectively [12]. From October 2021 to February/March 2022, another wave caused by Omicron (BA.1) occurred although less positivity and mortality were recorded [14]. Following this period, the disease became stable as vaccination and natural infection achieved herd immunity.

## 2.3 Study population and sample

The study population was maternal deaths in the study setting. The study sample comprised all women who died in the Klerksdorp/Tsphepong hospital complex that met the definition of maternal deaths during the 4-year study period. This was the eligibility criteria. According to the WHO, maternal mortality is the demise of any woman during the period of pregnancy (irrespective of the weeks of gestational age and location of the pregnancy) or within 42 days following childbirth as a result of causes related to or exaggerated by pregnancy but excluding accidental causes [15].

## 2.4 Inclusion and exclusion criteria

The inclusion criteria were all pregnant women and those in the puerperium (within 42 days of childbirth) who died in the Klerksdorp/Tshepong hospital complex. The exclusion criteria were (i) maternal deaths that occurred outside Klerksdorp/Tshepong hospital complex; and (ii) pregnant and postnatal women who died following accidental causes.

## 2.5 Data collection

The admission and discharge ward registers in the Obstetrics and Gynecology Department in the Klerksdorp/Tshepong hospital complex together with the departmental list of maternal deaths were used to identify the maternal deaths and childbirths that occurred in the hospital during the four-year period under review. The files of the listed deceased women were retrieved from the records department of the hospital, and the data collection and review of the clinical records were performed by the principal investigator. Afterwards, the research supervisors discussed the maternal deaths with the principal investigator during plenary meetings. Fig 1 is a flow diagram illustrating the data collection processes and the type of data collected.

## 2.6 Data analysis

Data was analyzed using SPSS version 29 (IBM, Armonk, NY, USA), with Joinpoint Regression Program, version 5.0.2 (National Cancer Institute https://surveillance.cancer.gov/joinpoint/) being used to evaluate the six-monthly percentage

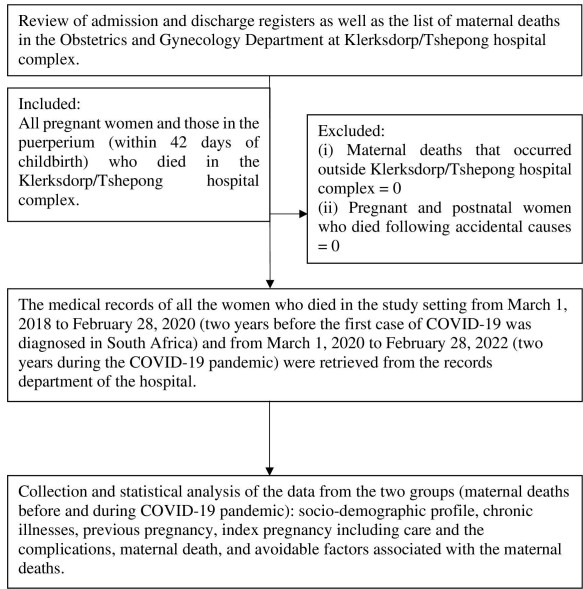

**Fig 1. Flow diagram on data collection.**

change in iMMR. A normality check was performed on numerical data using the skewness-Kurtosis test and visual assessment of the distribution of data in plots and charts including the normal curve on a histogram. Categorical variables were presented as frequencies and percentages. Numerical variables were summarized as mean with standard deviation (for normally distributed data) and median with interquartile range (for skewed data). Descriptive statistics was used to compute the trend in maternal deaths, live births, and iMMR. The iMMR was calculated by dividing the number of maternal deaths during a specified period by the total live births during the same period, multiplied by 100 000. A line graph showing six and 12-monthly trends in iMMR was plotted. The six-monthly point periods show the trend in iMMR and provide eight data points, which exceeds the minimum seven required for joinpoint regression analysis [16]. Numerical variables in the two periods (before and during the COVID-19 pandemic) were compared using the student t-test and Mann-Whitney U test for normally distributed data and skewed data, respectively. Categorical data were compared using Pearson's Chi-square distribution test and the Fisher exact test, as appropriate. Statistical significance was set at $p \leq 0.05$. Missing data were excluded from the calculation of the p-value.

## 2.7 Ethics

Approval from the University of Witwatersrand Human Research Ethics Committee (Reference M220359) was obtained before the inception of the study. The management of the Klerksdorp/Tshepong hospital complex also gave approval. During data collection, the medical records of the individual participants (deceased women) had their identities which were accessible to the principal investigator. However, no information that could identify individual participants was used after data collection (including during data capturing, statistical analysis, or reporting of the study findings).

## 3. Results

There were 58 maternal deaths during the study period, but the medical record of one woman who died during the COVID-19 pandemic could not be found. She was excluded from the analysis but included in the mortality trends.

### 3.1 Sociodemographic characteristics

There was no statistical difference between the two study periods in terms of the socio-demographic characteristics of the women (Table 1). All 58 reported maternal deaths were Black African women. There was no marked difference between the mean age (in years) of the two groups: 31.0±6.0 before and 31.1±6.2 during the COVID-19 pandemic (p=0.822). Two maternal deaths occurred among teenagers in the period before the COVID-19 pandemic.

### 3.2 Antenatal profile, COVID-19 status, and chronic medical conditions

Table 2 is a summary of the antenatal profile, COVID-19 status, and chronic medical condition of the women. Nearly all (94.7%) had been referred to the study setting. A total of 57.9% (33/57) of the women had antenatal care (18 before and 15 during the COVID-19 pandemic, p=0.056). The median parity of all the women was 3 (Interquartile range [IQR] 2–4), comprising 3 (2–3) before and 3 (2–4) during the COVID-19 pandemic, p=0.299. At booking, 57.9% (33/57) of the women had anemia: 62.7% (22/35) before and 50.0% (11/22) during the pandemic, p=0.476.

More than half of the women (66.7%, 38/57) had a pre-existing chronic medical condition before pregnancy with the most common being HIV infection (61.4%, 35/57). Of the women with HIV infection, the median CD4 count (cells/µl) was 312 (IQR 39–459), comprising 116 (11.5–378) before and 331 (197–667) during the COVID-19 pandemic, p=0.133. Of the 35 women with HIV infection 19 (54.3%) were on antiretroviral therapy: 12/35 before and 7/22 during the COVID-19 pandemic. The second most common chronic condition was chronic hypertension which occurred in 5.3.% (3/57) of all the women (two before and one during the COVID-19 pandemic).

**Table 1. Sociodemographic characteristics of the women.**

| Variable | Values: n (%), mean±SD | | | p-value |
|---|---|---|---|---|
| | **Before COVID-19 (March 2018 to February 2019)** | **During COVID-19 (March 2020 to February 2021)** | **Total** | |
| Mean age±SD (years) | 31.0±6.90 | 31.1±6.20 | 31.0±6.0 | 0.822[t] |
| Age (years), n (%) | | | | 0.344[f] |
| ≤24 | 3(8.6) | 5(22.7) | 8(14.0) | |
| 25–34 | 25(71.4) | 12(54.5) | 37(64.9) | |
| ≥35 | 7(20.0) | 5(22.7) | 12(21.1) | |
| Total | 35(100.0) | 22(100.0) | 57(100.0) | |
| Residence, n (%) | | | | 0.621[f] |
| Kenneth Kaunda District | 22(62.9) | 17(77.3) | 39(68.4) | |
| Outside Kenneth Kaunda district but in the same province | 12(34.3) | 5(22.7) | 17(29.8) | |
| Outside the province | 1(2.9) | 0(0.0) | 1(1.8) | |
| Total | 35(100.0) | 22(100.0) | 57(100.0) | |
| Marital status | | | | 0.652[f] |
| Single | 32(91.4) | 21(95.5) | 53(93.0) | |
| Married | 3(8.6) | 1(4.5) | 4(7.0) | |
| Total | 35(100.0) | 22(100.0) | 57(100.0) | |
| Occupation | | | | 1.00[f] |
| Employed | 2(5.7) | 2(9.1) | 4(7.0) | |
| Unemployed | 33(94.3) | 20(90.9) | 53(93.0) | |
| Total | 35(100.0) | 22(100.0) | 57(100.00) | |
| Race | | | | Constant |
| Black | 35(100.0) | 22(100.0) | 57(100.0) | – |
| Total | 35(100.0) | 22(100.0) | 57(100.0) | |
| Cigarette smoking | | | | 0.134[f] |
| Yes | 7(20) | 1(4.5) | 8(14.0) | |
| No | 28(80) | 21(95.5) | 49(86) | |
| Total | 35(100.0) | 22(100.0) | 57(100.0) | |
| Substance use | | | | 0.192[f] |
| Yes | 2(5.7) | 4(18.2) | 6(10.5) | |
| No | 33(94.3) | 18(81.8) | 51(89.5) | |
| Total | 35(100.0) | 22(100.0) | 57(100.0) | |
| District referred from | | | | 0.363[f] |
| Kenneth Kaunda | 24(68.6) | 16(72.7) | 40(70.2) | |
| Bojanala | 0(0.0) | 2(9.1) | 2(3.5) | |
| Ngaka Modiri Molema | 4(11.4) | 1(4.5) | 5(8.8) | |
| Ruth Segomotsi Mompati | 6(17.1%) | 3(13.6) | 9(15.8) | |
| Kuruman in Northern Cape | 1(2.9) | 0(0.0) | 1(1.8) | |
| Total | 35(100.0) | 22(100.0) | 57(100.0) | |

Abbreviations: t, student t-test; f, Fisher exact test. The p-value applies only to the comparison between the periods before and during COVID-19)

**Table 2. Antenatal profile, COVID-19 status, and chronic medical conditions.**

| Variable | Values: n (%), median [IQR] | | | p-value |
|---|---|---|---|---|
| | Before COVID-19 (March 2018 – February 2019) | During COVID-19 (March 2020 – February 2021) | Total | |
| Referral status at the study setting | | | | 1.00[f] |
| Referred | 33(94.3) | 21(95.5) | 54(94.7) | |
| Not referred | 2(5.7) | 1(4.5) | 3(5.3) | |
| Total | 35(100.0) | 22(100.0) | 57(100.0) | |
| Delay in referral | | | | 1.00[f] |
| Yes | 7(20.0) | 4(18.2) | 11(19.3) | |
| No | 28(80.0) | 18(81.8) | 46(80.7) | |
| Total | 35(100.0) | 22(100.0) | 57(100.0) | |
| Referring institution | | | | 0.132[f] |
| Clinic in Kenneth Kaunda | 16(45.7) | 10(45.5) | 26(45.6) | |
| Hospital in Kenneth Kaunda | 1(2.1) | 5(22.7) | 6(10.5) | |
| Institution outside Kenneth | | | | |
| Kaunda in North West Province | 11(31.4) | 6(27.3) | 17(29.8) | |
| Institution outside North | 2(5.7) | 0(0.0) | 2(3.5) | |
| Tshepong Hospital | 3(8.6) | 0(0.0) | 3(5.3) | |
| Not referred | 2((5.7) | 1(4.5) | 3(5.3) | |
| Total | 35(100.0) | 22(100.0) | 57(100.0) | |
| Type of chronic condition, n (%) | | | | 0.810[f] |
| HIV | 20(57.1) | 14(63.6) | 34(59.6) | |
| Chronic Hypertension | 1(2.9) | 1(4.5) | 2(3.5) | |
| Epilepsy | 0 | 1(4.5) | 1(1.8) | |
| HIV and hypertension | 1(2.9) | 0 | 1(1.8) | |
| No chronic condition | 12(34.3) | 6(27.3) | 18(31.6) | |
| Missing data | 1(2.9) | 0 | 1(1.8) | |
| Total | 35(100.0) | 22(100.0) | 57(100.0) | |
| HIV antiretroviral therapy(ART) | | | | 1.00[f] |
| On ART | 12(34.3) | 7(31.8) | 19(33.3) | |
| Not ART | 6(17.1) | 3(13.6) | 9(25.7) | |
| HIV negative | 14(24.6) | 8(36.4) | 22(62.8) | |
| Missing data among HIV women | 3(8.6) | 4(18.2) | 7(12.3) | |
| Total | 35(100.0) | 22(100.0) | 57(100.0) | |
| CD4 count (cells/μl) range and median, n = 26 | | | | 0.133[m] |
| Minimum | 4 | 46 | 4 | |
| Maximum | 1484 | 1067 | 1484 | |
| Median, IQR | 116[11.5–378] | 331[197–667] | 312 [39–459] | |
| Variable | Values: n (%), median [IQR] | | | p-value |
| | Before COVID-19 (March 2018 – February 2019) | During COVID-19 (March 2020 – February 2021) | Total | |
| CD4 Category (cells/μl) | | | | 0.319[f] |
| <199 | 9(42.9) | 2(14.3) | 11(31.4) | |
| 200-499 | 5(23.8) | 5(35.7) | 10(28.6) | |
| ≥500 | 3(14.3) | 2(14.3) | 5(14.3) | |
| Missing data | 4(19.0) | 5(35.7) | 9(25.7) | |
| Total | 21(100.0) | 14(100.0) | 35(100.0) | |

*(Continued)*

| Variable | Values: n (%), median [IQR] | | | p-value |
|---|---|---|---|---|
| | **Before COVID-19 (March 2018 – February 2019)** | **During COVID-19 (March 2020 – February 2021)** | **Total** | |
| Bactrim use if CD4 < 200 cells/μl | | | | 0.616f |
| Yes | 2(9.5) | 0(0.00) | 2(5.7) | |
| No | 10(47.6) | 7(50.0) | 17(48.6) | |
| Unknown | 1(4.8) | 0(0.00) | 1(2.9) | |
| CD4 ≥ 200 | 8(38.1) | 7(50.0) | 15(42.9) | |
| Total | 21(100.0) | 14(100.0) | 35(100.0) | |
| HIV viral load (copies/ml), range, and median | | | | 0.285m |
| Minimum | 0 | 0 | 0 | |
| Maximum | 4697585 | 1180739 | 4697585 | |
| Median (IQR) | 1787 [45–83010] | 16844.5 [1458–171120.8] | 4738 [60.5 −106274] | |
| HIV viral load (copies/ml), category | | | | 0.261m |
| <200 | 6(28.6) | 2(14.3) | 8(22.9) | |
| ≥200–99999 | 7(33.3) | 4(28.6) | 11(31.4) | |
| ≥100000 | 2(9.5) | 4(28.6) | 6(17.1) | |
| Missing data | 6(28.6) | 4(28.6) | 10(28.6) | |
| Total | 21(100.0) | 14(100.0) | 35(100.0) | |
| Previous pregnancy ≥ 24 weeks' gestation | | | | 0.141m |
| Minimum | 1 | 1 | 1 | |
| Maximum | 9 | 6 | 9 | |
| Median [IQR] | 3[2-3] | 3[2-4] | 3[2-4] | |
| Previous pregnancy ≥ 24 weeks' gestation | | | | 0.299f |
| 1 | 5(14.2) | 2(9.1) | 7(12.3) | |
| 2–4 | 27(77.1) | 16(72.7) | 43(75.3) | |
| ≥5 | 2(5.7) | 4(18.1) | 6(10.6) | |
| Missing data | 1(2.9) | 0 | 1(1.8) | |
| Total | 35(100.0) | 22(100.0) | 57(100.0) | |
| Previous pregnancy less than 24 weeks' gestation | | | | 0.397m |
| Minimum | 0 | 0 | 0 | |
| Maximum | 8 | 5 | 8 | |
| Median [IQR] | 2[1-2] | 2[1-3] | 2[1-3] | |
| Variable | Values: n (%), median [IQR] | | | p-Value |
| | Before COVID-19 (March 2018 – February 2019) | During COVID-19 (March 2020 – February 2021) | Total | |
| Previous pregnancy less than 24 weeks' gestation | | | | 0.326f |
| 0 | 4(11.4) | 1(4.5) | 5(8.8) | |
| 1-2 | 23(65.7) | 13(59.1) | 36(63.1) | |
| ≥3 | 7(20.0) | 8(36.4) | 15(26.3) | |
| Missing data | 1(2.9) | 0 | 1(1.8) | |
| Total | 35(100.0) | 22(100.0) | 57(100.0) | |
| Previous termination of pregnancy | | | | 0.652m |
| Minimum | 0 | 0 | 0 | |
| Maximum | 1 | 1 | 1 | |
| Median [IQR] | 0[0−0] | 0[0−0] | 0[0−0] | |
| Previous termination of pregnancy | | | | |

*(Continued)*

| Variable | Values: n (%), median [IQR] | | | *p*-value |
|---|---|---|---|---|
| | **Before COVID-19 (March 2018 – February 2019)** | **During COVID-19 (March 2020 – February 2021)** | **Total** | |
| 0 | 32(91.4) | 20(90.9) | 52(91.2) | 1.000[f] |
| 1 | 2(5.7) | 2(9.1) | 4(7.0) | |
| Missing data | 1(2.9) | 0 | 1(1.8) | |
| Total | 35(100.0) | 22(100.0) | 57(100.0) | |
| Previous caesarean section | | | | 0.698[f] |
| Yes | 6(17.1) | 6(27.3) | 12(21.0) | |
| No | 28(80.0) | 16(72.7) | 44(77.2) | |
| Unknown | 1(2.9) | 0(0.00) | 1(1.8) | |
| Total | 35(100.0) | 22(100.0) | 57(100.0) | |
| Attended antenatal care clinic | | | | 0.056[f] |
| Yes | 18(51.4) | 15(68.2) | 33(57.9) | |
| No | 13(37.1) | 2(9.1) | 15(26.3) | |
| Unknown | 3(8.57) | 4(18.2) | 7(12.3) | |
| Missing data | 1(2.9) | 1(4.5) | 2(3.5) | |
| Total | 35(100.0) | 22(100.0) | 57(100.0) | |
| Gestational age at booking | | | | 0.301[c] |
| Less than 20 weeks | 9(25.7) | 8(36.4) | 17(29.8) | |
| Above 20 weeks | 10(28.6) | 8(36.4) | 18(31.6) | |
| Unbooked | 13(37.1) | 4(18.2) | 17(29.8) | |
| Unspecified | 3(8.6) | 2(9.1) | 5(8.8) | |
| Total | 35(100.0) | 22(100.0) | 57(100.0) | |
| Anaemia during pregnancy | | | | 0.476[c] |
| Yes | 22(62.9) | 11(50.0) | 33(57.9) | |
| No | 9(25.7) | 7(31.8) | 16(28.1) | |
| Missing data | 4(11.4) | 4(18.2) | 8(14.0) | |
| Total | 35(100.0) | 22(100.0) | 57(100.0) | |
| Rapid Plasma Reagin | | | | 0.098[f] |
| Non-reactive | 21(60) | 16(72.7) | 37(65.0) | |
| Unknown | 11(31.4) | 2(9.1) | 13(22.7) | |
| Missing data | 3(8.6) | 4(18.2) | 7(12.3) | |
| Total | 35(100.0) | 22(100.0) | 57(100.0) | |
| Variable | Values: n (%), median [IQR] | | | *p*-value |
| | Before COVID-19 (March 2018 – February 2019) | During COVID-19 (March 2020 – February 2021) | Total | |
| Booking weight (kg), range and median | | | | 0.580[m] |
| Minimum | 43 | 36 | 36 | |
| Maximum | 563 | 117 | 563 | |
| Median (IQR) | 60[55–75] | 76[47.75–93.5] | 54.5[53 - 85.3] | |
| Booking weight (kg) | | | | 0.378[f] |
| <90 | 14(40) | 10(45.5) | 24(42.1) | |
| ≥90 | 2(5.7) | 4(18.2) | 6(10.6) | |
| Missing data | 19(54.3) | 8(36.4) | 27(47.3) | |
| Total | 35(100.0) | 22(100.0) | 57(100.0) | |
| Mid Upper Arm Circumference (cm), range, and median | | | | 0.790[m] |
| Minimum | 21 | 20 | 20 | |

*(Continued)*

Table 2. (Continued)

| Variable | Values: n (%), median [IQR] | | | p-value |
|---|---|---|---|---|
| | Before COVID-19 (March 2018 – February 2019) | During COVID-19 (March 2020 – February 2021) | Total | |
| Maximum | 245 | 39 | 245 | |
| Median [IQR] | 25[23–29.75] | 24.5[21–35] | 25[22.4 - 30.8] | |
| Mid Upper Arm Circumference (cm), range, and median | | | | |
| ≤22 | 3(8.6) | 5(22.7) | 8(14.0) | 0.101[f] |
| 23-33 | 11(31.4) | 5(22.7) | 16(28.1) | |
| ≥34 | 1(2.9) | 4(18.2) | 5(8.8) | |
| Missing data | 20(63.6) | 8(36.4) | 28(49.1) | |
| Total | 35(100.0) | 22(100.0) | 57(100.0) | |
| Height (cm) | | | | 0.505 |
| Minimum | 151 | 143 | 8 | |
| Maximum | 164 | 185 | 16 | |
| Median [IQR] | 155[152-162] | 159.5[151-163] | 5 | |
| Height (cm) | | | | |
| ≤150 | 3(8.6) | 2(9.1) | 5(8.8) | 1.000[f] |
| ≥151 | 12(34.3) | 12(54.5) | 24(42.1) | |
| Missing data | 20(57.1) | 8(36.4) | 28(49.1) | |
| Total | 35(100.0) | 22(100.0) | 57(100.0) | |
| COVID-19 | | | | <0.001[f] |
| Yes | 0(0.0) | 9(40.9) | 9(15.8) | |
| No | 35(100.0) | 13(59.1) | 48(84.2) | |
| Total | 35(100.0) | 22(100.0) | 57(100.0) | |
| Antenatal complications | | | | 0.906[c] |
| Present | 15(42.9) | 10(45.5) | 25(43.9) | |
| Absent | 14(40.0) | 10(45.5) | 24(42.1) | |
| Missing data | 6(17.1) | 2(9.1) | 8(14.0) | |
| Total | 35(100.0) | 22(100.0) | 57(100.0) | |
| Variable | Values: n (%), median [IQR] | | | p-value |
| | Before COVID-19 (March 2018 – February 2019) | During COVID-19 (March 2020 – February 2021) | Total | |
| Antenatal complication types | | | | 0.107[f] |
| Unbooked | 13(37.1) | 4(18.2) | 17(29.8) | |
| Preterm labour | 1(2.9) | 0(0.00) | 1(1.8) | |
| Genital warts | 1(2.9) | 0(0.00) | 1(1.8) | |
| Hypertension in pregnancy | 5(14.3) | 3(13.6) | 8(14.0) | |
| Lower respiratory infection | 3(8.6) | 0(0.00) | 3(5.3) | |
| *COVID-19 | 0(0.00) | 4(18.2) | 4(7.0) | |
| **Others | 4(11.4) | 4(18.2) | 8(14.0) | |
| Missing data | 8(22.8) | 7(31.8) | 15(26.3) | |
| Total | 35(100.0) | 22(100.0) | 57(100.0) | |
| Management of antenatal complications met expectation | | | | 0.545[f] |
| Yes | 7(20.0) | 4(18.2) | 11(19.3) | |
| No | 2(5.7) | 4(18.2) | 6(10.6) | |
| Sometimes | 6(17.1) | 4(18.2) | 10(17.5) | |
| Missing data | 20(57.1) | 10(45.5) | 30(52.6) | |

*(Continued)*

**Table 2.** (Continued)

| Variable | Values: n (%), median [IQR] | | | p-value |
|---|---|---|---|---|
| | Before COVID-19 (March 2018 – February 2019) | During COVID-19 (March 2020 – February 2021) | Total | |
| Total | 35(100.0) | 22(100.0) | 57(100.0) | |

*Of the 9 women diagnosed with COVID -19, a total of 4 were diagnosed during the antenatal period.

**Others were women who had more than one complication. Anemia was defined as a hemoglobin concentration less than 11g/ dl in the first trimester.
Abbreviations: c, Chi-square test; f, Fisher exact test; m, Mann-Whitney U test.

The most frequent antenatal complication that developed after booking was hypertensive disorders of pregnancy (HDP) and this occurred in 14.3% (5/35) and 13.6% (3/22) of the women before and during the COVID-19 pandemic, respectively. The second most common antenatal complication was COVID-19, which occurred in 7.0% (4/57) of all the women but, understandably, there was none before the COVID-19 period. Overall, 9 (15.8%) women had COVID-19 in total (Table 2). Except for COVID-19, there was no statistical difference between the two study periods in terms of the antenatal profile, COVID-19 status, and chronic medical condition (Table 2).

### 3.3. Trends in maternal deaths, births, and maternal mortality ratio

The trends in maternal deaths and live births during the four-year study period are summarized in Table 3. Of the 58 maternal deaths, 60.34% (35/58) occurred before and 39.66% (23/58) during the COVID-19 pandemic. The live births increased from 5 166 in the first year of the study, reaching a peak of 5 947 in the third year (March 2020 to February 2021) and thereafter declining to 5 476 in the fourth year. There was also a steady decline in maternal deaths from 17 to 11 during the four-year period. Similarly, the iMMR decreased steadily from 329.07 to 200.88 deaths per 100 000 live births (Fig 2).

Table 4 shows the six-monthly maternal deaths, deliveries, and iMMR across 8-point periods. The joinpoint regression analysis of the six-monthly iMMR showed a statistically significant declining trend of 9.59% between February 2018 – August 2018 and September 2021 – February 2022 (Fig 3). The six-monthly trends in iMMR are shown in Fig 4.

### 3.4 Intrapartum and postpartum outcomes and their complications

There was no statistically significant difference between the two periods in terms of intrapartum and postpartum outcomes and their complications (Table 5). Nonetheless, 40.4% (23/57) of the women delivered vaginally (16/35 before and 7/22 during the COVID-19 pandemic) compared to 33.3% (19/57) who delivered via caesarean delivery (9/35 before and 10/22 during COVID-19 pandemic). A respective total of 21% (12/57) and 5.3% (3/57) were undelivered and had a laparotomy for ectopic pregnancy. The most common indication for caesarean delivery was fetal distress which occurred in 12.2% (7/57) of the women with 4/35 before and 3/22 during the COVID-19 pandemic.

**Table 3. Twelve-monthly trends in maternal deaths, live births, and maternal mortality ratio.**

| Variables | Before COVID-19 | | During COVID-19 | |
|---|---|---|---|---|
| | Mar 2018 – Feb 2019 | Mar 2019 – Feb 2020 | Mar 2020 – Feb 2021 | Mar 2021 – Feb 2022 |
| Maternal deaths | 17 | 18 | 12 | 11 |
| Live births | 5166 | 5866 | 5947 | 5476 |
| Total births | 5333 | 6070 | 6153 | 5647 |
| Maternal mortality ratio per 100 000 live births | 329.1 | 306.9 | 201.8 | 200.9 |

Overall, the maternal mortality ratio per 100 000 live births was 317.3 before COVID-19, 201.3 during COVID-19, and 258.3 during the 4-year period of the study.

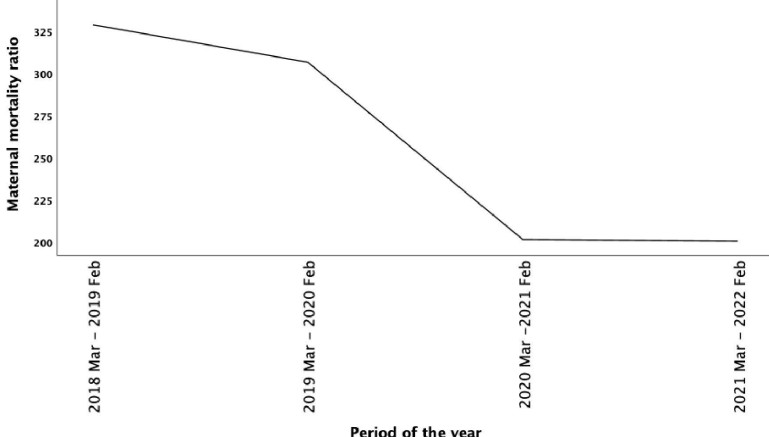

**Fig 2. Twelve-monthly trends in maternal mortality ratio.** Abbreviations: Feb, February; Mar, March.

**Table 4. Six-monthly maternal deaths, deliveries, births, and maternal mortality ratio.**

| Period | Maternal deaths | Total deliveries | Total births | Total live births | Maternal mortality ratio per 100 000 live births |
|---|---|---|---|---|---|
| Before COVID period | | | | | |
| Feb 2018 – Aug 2018 | 10 | 2685 | 2720 | 2632 | 379.9 |
| Sep 2018 – Feb 2019 | 7 | 2571 | 2613 | 2534 | 276.2 |
| Mar 2019 – Aug 2019 | 11 | 2942 | 2979 | 2884 | 381.4 |
| Sep 2019 – Feb 2020 | 7 | 2956 | 3091 | 2982 | 234.7 |
| During COVID-19 period | | | | | |
| Mar 2020 – Aug 2020 | 6 | 3126 | 3188 | 3090 | 194.2 |
| Sep 2020 – Feb 2021 | 6 | 2920 | 2965 | 2857 | 210.0 |
| Mar 2021 – Aug 2021 | 6 | 2865 | 2906 | 2816 | 213.1 |
| Sep 2021 – Feb 2022 | 5 | 2689 | 2741 | 2760 | 181.2 |

The median APGAR score for the two periods was 8 (0–10) comprising 7.5 (0–9) before and 9 (6.25–10) during the COVID-19 pandemic, p = 0.116. The median birth weight (g) for the babies of the women in the two periods was 2 130 (1 300–2 750) comprising 1 850 (1 187.5–2 625) before and 2 400 (1 395–3 000) during COVID-19 pandemic, p = 0.226. The most common postpartum complications before the COVID-19 pandemic were puerperal sepsis 14.3% (5/35), postpartum hemorrhage (PPH) 11.4 (4/35), and preeclampsia with severe features 11.4% (4/35). During the COVID-19 pandemic, the most common postpartum complications were COVID-19 13.6% (3/22), preeclampsia with severe features 13.6% (3/22), and postpartum haemorrhage (PPH) 9.1% (2/22).

### 3.5 Critical care admissions and mechanical ventilation

Table 6 shows critical care admissions and mechanical ventilation. A higher percentage of women, 38.1% (8/22), were admitted to the high care unit during the COVID-19 pandemic compared to 31.4% (11/35) before the pandemic. Before the COVID-19 pandemic, sepsis was the most common indication for high-care admission, occurring in 14.3% (5/35) of the women. However, during the COVID-19 pandemic, the most common indications for high-care admission were pre-eclampsia with severe features 9.1% (2/22) and cardiac diseases 9.1% (2/22).

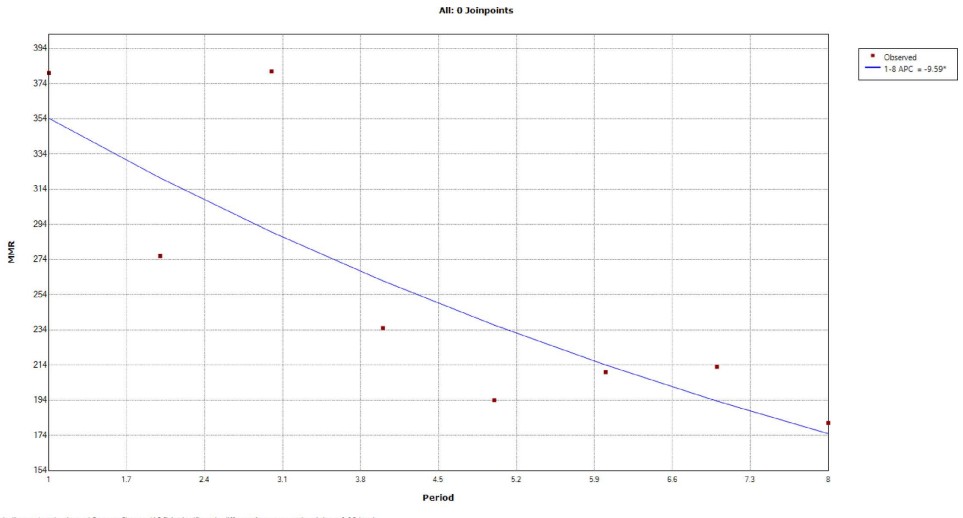

**Fig 3. Joinpoint regression analysis of the six-monthly maternal mortality ratio.** Explanation: The Annual percentage change (APC) denotes six monthly percentage change. There were 8 time periods of six months each: 1 – 8 (square dots), consisting of 1 = Feb 2018 – Aug 2018; 2 = Sep 2018 – Feb 2019; 3 = Mar 2019 – Aug 2019; 4 = Sep 2019 – Feb 2020; 5 = Mar 2020 – Aug 2020; 6 = Sep 2020 – Feb 2021; 7 = Mar 2021 – Aug 2021; and 8 = Sep 2021 – Feb 2022.

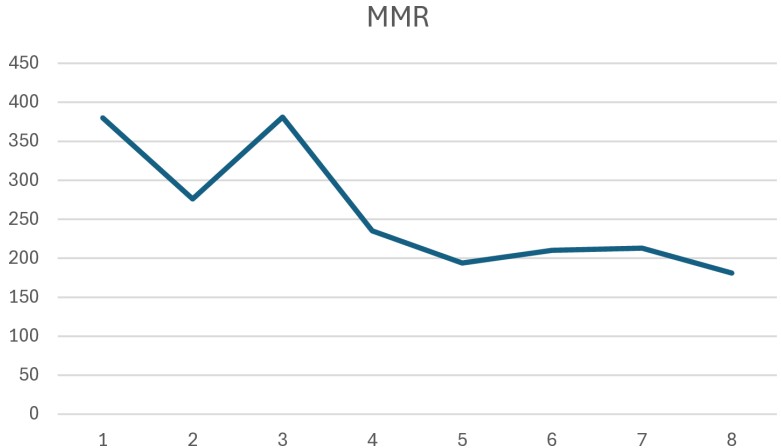

**Fig 4. Six-monthly trends in maternal mortality ratio.** Explanation: Abbreviation: MMR , Maternal mortality ratio. There were 8 time periods 1 – 8, consisting of 1 = Feb 2018 – Aug 2018; 2 = Sep 2018 – Feb 2019; 3 = Mar 2019 – Aug 2019; 4 = Sep 2019 – Feb 2020; 5 = Mar 2020 – Aug 2020; 6 = Sep 2020 – Feb 2021; 7 = Mar 2021 – Aug 2021; and 8 = Sep 2021 – Feb 2022.

The proportion of women admitted to the intensive care unit (ICU) was 71.4% (25/35) before and 68.2% (15/22) during the COVID-19 pandemic. The most common indications for ICU admission were acute respiratory distress at 34.3% (12/35) before and 36.4% (8/22) during the COVID-19 pandemic, $p = 0.151$. The rate of mechanical ventilation was 74.3% (26/35) before and 52.38% (11/22) during the COVID-19 pandemic ($p = 0.026$).

### 3.6  Time *of* hospital admission and hour *of* maternal death

Table 7 shows the time of hospital admission and hour of maternal deaths, with no statistically significant difference between the periods before and during the COVID-19 pandemic, $p > 0.05$. Most hospital admissions occurred during the

**Table 5. Intrapartum and postpartum outcomes and their complications.**

| Variable | Values: n (%), median [IQR] | | | p-value |
|---|---|---|---|---|
| | Before COVID (January 2018 – December 2019) | During COVID (January 2020 – December 2021) | Total | |
| Mode of delivery n (%) | | | | 0.266[f] |
| Vaginal birth | 16(45.7) | 7(31.8) | 23(40.4) | |
| Caesarean delivery | 9(25.7) | 10(45.5) | 19(33.3) | |
| Undelivered | 7(20.0) | 5(22.7) | 12(21.0) | |
| Laparotomy for ectopic pregnancy | 3(8.6) | 0(0.00) | 3(5.3) | |
| Total | 35(100.0) | 22(100.0) | 57(100.0) | |
| | | | | |
| Indication for caesarean section, n (%) | | | | 0.367[f] |
| Fetal distress | 4(11.4) | 3(13.6) | 7(12.2) | |
| Previous uterine surgery | 0(0.0) | 3(13.6) | 3(5.3) | |
| Preeclampsia with severe Features | 3(8.6) | 1(4.5) | 4(7.0) | |
| Maternal resuscitation | 2(5.7) | 1(4.5) | 3(5.3) | |
| Breech presentation | 0(0.00) | 1(4.5) | 1(1.8) | |
| Severe COVID-19 Pneumonia | 0(0.00) | 1(4.5) | 1(1.8) | |
| Had vaginal delivery | 16(45.7) | 7(31.8) | 23(40.4) | |
| Missing data | 10(28.6) | 5(22.7) | 15(26.3) | |
| Total | 35(100.0) | 22(100.0) | 57(100.0) | |
| | | | | |
| Blood loss (ml) during childbirth | | | | 0.845[m] |
| Minimum | 50 | 200 | 50 | |
| Maximum | 2000 | 1100 | 2000 | |
| Median [IQR] | 500[200 - 950] | 500[250 - 1000] | 500[275 - 1000] | |
| | | | | |
| Blood loss (ml) during childbirth, n (%) | | | | 0.413[f] |
| ≤499 | 6(17.1) | 3(13.6) | 9 (15.8) | |
| 500–999 | 4(11.4) | 3(13.6) | 7(12.3) | |
| 1000–1499.9 | 1(2.9) | 3(13.6) | 4(7.0) | |
| ≥1500 | 2(5.7) | 0(0.00) | 2(3.5) | |
| Missing data | 22(62.9) | 13(59.1) | 35(61.4) | |
| Total | 35(100.0) | 22(100.0) | 22(100.0) | |
| | | | | |
| Apgar score | | | | 0.116[m] |
| Minimum | 0 | 0 | 0 | |
| Maximum | 10 | 10 | 10 | |
| Median (IQR) | 7.5[0 - 9] | 9 [6.25 - 10] | 8[0 - 10] | |
| Variable | Values: n (%), median [IQR] | | | p-value |
| | Before COVID (January 2018 – December 2019) | During COVID (January 2020 – December 2021) | Total | |
| Apgar score category, n (%) | | | | |
| Stillbirth (0) | 0(0.0) | 0(0.0) | 0(0.0) | Constant |
| Asphyxia (1–6) | 0(0.0) | 0(0.0) | 0(0.0) | |
| Normal Apgar (≥7) | 15((42.9) | 14(63.6) | 29 (50.8) | |
| Undelivered | 7(20) | 5(22.7) | 12(21.0) | |
| Ectopic laparotomy | 3(8.6) | 0(0.0) | 3(5.3) | |

*(Continued)*

| Variable | Values: n (%), median [IQR] | | | p-value |
|---|---|---|---|---|
| | Before COVID (January 2018 – December 2019) | During COVID (January 2020 – December 2021) | Total | |
| Missing data | 10(28.5) | 3(13.6) | 13(22.8) | |
| Total | 35(100.0) | 22(100.0) | 57(100.0) | |
| | | | | |
| Birth weight (g) | | | | 0.226m |
| Minimum | 450 | 890 | 450 | |
| Maximum | 3400 | 3400 | 3400 | |
| Median, IQR | 1850[1187.5–2625] | 2400[1395–3000] | 2130[1300–2750] | |
| | | | | |
| Birth weight (gram) category, n (%) | | | | 0.563f |
| ≤999g | 4(11.4) | 1(4.5) | 5(8.8) | |
| 1000-2499 g | 8(22.9) | 6(27.3) | 14(24.6) | |
| ≥2500 g | 6(17.1) | 6(27.3) | 12(21.0) | |
| Missing data | 17(48.6) | 9(40.9) | 26(45.6) | |
| Total | 35(100.0) | 22(100.0) | 57(100.0) | |
| | | | | |
| Gender of baby, n (%) | | | | 0.755c |
| Male | 10(28.6) | 7(31.8) | 17(29.8) | |
| Female | 8(22.9) | 7(31.8) | 15(26.3) | |
| Undelivered | 7(20.0) | 5(22.7) | 12(21.0) | |
| Ectopic pregnancy | 3(5.3) | 0(0.0) | 3(5.3) | |
| Missing data | 7(20.0) | 3(5.3) | 10(28.6) | |
| Total | 35(100.0) | 22(100.0) | 57(100.0) | |
| | | | | |
| Intrapartum complications, n (%) | | | | 0.013f |
| Yes | 0 | 4(18.2) | 4(7.0) | |
| No | 26(74.3) | 11(50.0) | 37(64.9) | |
| Missing data | 9(25.7) | 7(31.8) | 16(28.1) | |
| | | | | |
| Type of intrapartum complications, n (%) | | | | Constant |
| Eclampsia | 0(0.00) | 1(4.5) | 1(1.8) | |
| Abruption placenta | 0(0.00) | 1(4.5) | 1(1.8) | |
| Fetal Distress | 0(0.00) | 1(4.5) | 1(1.8) | |
| Cephalo-pelvic disproportion | 0(0.00) | 1(4.5) | 1(1.8) | |
| No complication | 26(74.3) | 11(50.0) | 37(64.9) | |
| Missing data | 9(24.7) | 7(32.0) | 16(28.0) | |
| Total | 35(100.0) | 22(100.0) | 57(100.0) | |
| Variable | Values: n (%), median [IQR] | | | p-value |
| | Before COVID (January 2018 – December 2019) | During COVID (January 2020 – December 2021) | Total | |
| Postpartum complications, n (%) | | | | 0.859c |
| Yes | 17(48.6) | 12(54.6) | 29(50.9) | |
| No | 8(22.9) | 5(22.7) | 13(22.8) | |
| Missing data | 10(28.6) | 5(22.7) | 15(26.3) | |
| Total | 35(100.0) | 22(100.0) | 57(100.0) | |

*(Continued)*

**Table 5.** (Continued)

| Variable | Values: n (%), median [IQR] | | | p-value |
|---|---|---|---|---|
| | **Before COVID (January 2018 – December 2019)** | **During COVID (January 2020 – December 2021)** | **Total** | |
| | | | | |
| Type of postpartum complication, n (%) | | | | 0.349[f] |
| Postpartum hemorrhage | 4(11.4) | 2(9.1) | 6(10.5) | |
| Puerperal Sepsis | 5(14.3) | 1(4.5) | 6(10.5) | |
| Venous thromboembolism | 1(2.9) | 2(9.1) | 3(5.3) | |
| Cardiac arrest during caesarean delivery | 2(5.7) | 1(4.5) | 3(5.3) | |
| Preeclampsia with severe features | 4(11.4) | 3(13.6) | 7(12.3) | |
| Perforated Uterus | 1(2.9) | 0(0.00) | 1(1.8) | |
| COVID-19 | 0(0.00) | 3(13.6) | 3(5.3) | |
| No complication | 18(51.5) | 10(45.6) | 28(49.1) | |
| Total | 35(100.0) | 22(100.0) | 57(100.0) | |

Abbreviations: c, Chi-square test; f, Fisher exact test; m, Mann-Whitney U test

day. However, the majority of maternal deaths occurred during the night: 74.3% (26/35) before and 59.1% (13/22) during the COVID-19 pandemic, p = 0.494. Most of the maternal deaths occurred outside the Obstetrics and Gynaecology department, 81.8% before and 87.7% during the COVID-19 pandemic.

### 3.7 Primary and final causes *of* maternal deaths

Table 8 shows the primary and final causes of maternal deaths. Preeclampsia with severe features (22.9%) was the most common primary cause of maternal deaths before the COVID-19 pandemic, with tuberculosis being second (20%), early pregnancy bleeding third (17.2%), and sepsis fourth (14.3%). Conversely, COVID-19 (31.8%), and tuberculosis (13.6%) were the first and second most common primary causes of maternal deaths during the COVID-19 pandemic, with the third most common cause being four conditions namely PPH, pulmonary embolism, sepsis and preeclampsia with severe features, each having a percentage occurrence of 9.1%. The primary causes of maternal deaths for the two periods (p = 0.009), however, were statistically significantly different.

The most common final causes of maternal deaths before and during the COVID-19 pandemic were multiple organ failure (45.7%) and cardiopulmonary arrest (50%); however, there was no statistically significant difference in the final causes of maternal deaths between the two periods (p = 0.092).

### 3.8 Avoidable factors

The patient-, healthcare professional- and administrative-related AVFs associated with the maternal deaths are presented in Table 9. Each woman had at least one AVF that was associated with her management. Before the COVID-19 pandemic, patient-, healthcare professional-, and administrative-related AVFs occurred in 62.9% (22/35), 62.9% (22/35), and 57.1% (20/35) of the women, respectively. Before the COVID-19 pandemic, the most common patient-, healthcare professional- and administrative-related AVFs were not having any antenatal care, wrong treatment, and lack of ICU/high care spaces respectively.

During the COVID-19 pandemic, patient-, healthcare professional- and administrative-related AVFs occurred in 36.4% (8/22), 54.5% (12/22), and 50% (11/22) of the women, respectively. In the same period, the most common patient-,

healthcare professional- and administrative-related AVFs were delay in seeking medical treatment, lack of critical care skills, and unavailability of ICU/high care beds, in this order.

Table 10 shows the AVFs associated with the management of causes of the maternal deaths. The conditions associated with the greatest number of AVFs were pregnancy-related infection and HDP. The most common AVF associated with non-pregnancy-related infections was poor compliance with antiretroviral drugs: with frequencies of 12 before and eight

**Table 6. Critical care admissions and mechanical ventilation.**

| Variable | Value: n (%) | | | p-value |
|---|---|---|---|---|
| | Before COVID-19 | During COVID-19 | Total | |
| Admission to high care unit, n (%) | | | | 0.700c |
| Yes | 11(31.4) | 8(38.10) | 19(33.3) | |
| No | 24(68.6) | 14(63.6) | 38(66.7) | |
| Total | 35(100.0) | 22(100.0) | 57(100.0) | |
| | | | | |
| Indications for admission to high care unit, n (%) | | | | 0.418f |
| Preeclampsia with severe features | 2(5.7) | 2(9.1) | 4(7.0) | |
| Postpartum haemorrhage | 1(2.9) | 1(4.5) | 2(3.5) | |
| Sepsis | 5(14.3) | 1(4.5) | 6(10.5) | |
| Cardiac diseases including supraventricular tachycardia | 3(8.6) | 2(9.1) | 5(8.8) | |
| Other indications | 0(00) | 2(9.1) | 2(3.5) | |
| | | | | |
| Straight admission to ICU | 24(68.6) | 14(63.6) | 38(66.7) | |
| Total | 35(100.0) | 22(100.0) | 57(100.0) | |
| | | | | |
| Admission to ICU, n (%) | | | | 1.000c |
| Yes | 25(71.4) | 15(68.2) | 40(70.2) | |
| No | 10(28.6) | 6(27.2) | 16(28.0) | |
| Missing data | 0 (0) | 1(4.5) | 1(1.8) | |
| Total | 35(100.0) | 22(100.0) | 57(100.0) | |
| | | | | |
| Indication for admission to ICU, n (%) | | | | 0.151f |
| Acute respiratory distress | 12(34.3) | 8(36.4) | 20(35.1) | |
| GCS of 7 and less | 3(8.6) | 3(13.6) | 6(10.5) | |
| Inotropic support | 9(25.7) | 2(9.1) | 11(19.3) | |
| Cardioversion | 2(5.7) | 0(0.00) | 2(3.5) | |
| Fluid resuscitation | 0(0.00) | 2(9.1) | 2(3.5) | |
| Not admitted to ICU | 9(25.7) | 7(31.8) | 16(28.1) | |
| Total | 35(100.0) | 22(100.0) | 57(100.0) | |
| | | | | |
| Mechanical ventilation, n (%) | | | | 0.026c |
| Yes | 26(74.3) | 11(52.38) | 37(64.9) | |
| No | 7(20.0) | 11(47.62) | 18(31.6) | |
| Missing data | 2(5.7) | 0(00) | 2(3.5) | |
| Total | 35(100.0) | 22(100.0) | 57(100.0) | |

*Resuscitation describes women who were admitted to the intensive care unit (ICU) to control central venous pressure.

Abbreviations: GCS, Glasgow Coma Scale. c: Chi-square test; f: Fisher exact test.

**Table 7. Time of hospital admission and maternal death.**

| Variable | Value: n (%) | | | p-value |
|---|---|---|---|---|
| | Before COVID-19 | During COVID-19 | Total | |
| Time of hospital admission, n (%) | | | | 0.192[f] |
| Day (08:00–15:59 hours) | 20(57.1) | 7(31.8) | 27(47.4) | |
| Evening (16:00–18:59 hours) | 6(17.1) | 5(22.7) | 11(19.3) | |
| Night (19:00–07:59 hours) | 9(25.7) | 10(45.5) | 19 (33.3) | |
| Total | 35(100.0) | 22(100.0) | 57(100.0) | |
| | | | | |
| Time of death, n (%) | | | | 0.494[f] |
| Day (08:00–15:59 hours) | 5(14.3) | 5(22.7) | 10(17.5) | |
| Evening (16:00–18:59 hours) | 4(11.4) | 4(18.2) | 8(14.1) | |
| Night (19:00–07:59 hours) | 26(74.3) | 13(59.1) | 39(68.4) | |
| Total | 35(100.0) | 22(100.0) | 57(100.0) | |
| | | | | |
| Department where death occurred, n (%) | | | | 0.411[f] |
| Within Obstetrics and Gynaecology department | 3(8.6) | 4(18.2) | 7(12.3) | |
| Outside Obstetrics and Gynaecology department but including ICU setting | 32(91.4) | 18(81.8) | 50(87.7) | |
| Total | 35(100.0) | 22(100.0) | 57(100.0) | |

Abbreviations: c, Chi-square test; f, Fisher exact test; ICU, intensive care unit

**Table 8. Causes of maternal deaths.**

| Variable | Value: n (%) | | | p-value |
|---|---|---|---|---|
| | Before COVID (January 2018 – December 2019) | During COVID (January 2020 – December 2021) | Total | |
| Primary causes of death | | | | 0.009[f] |
| Postpartum haemorrhage | 2(5.7) | 2(9.1) | 4(7.0) | |
| Pulmonary embolism | 2(5.7) | 2(9.1) | 4(7.0) | |
| Tuberculosis | 7(20) | 3(13.6) | 10(17.5) | |
| Sepsis | 5(14.3) | 2(9.1) | 7(12.3) | |
| Early pregnancy bleeding | 6(17.2) | 0(0.00) | 6(10.5) | |
| Preeclampsia with severe features | 8(22.9) | 2(9.1) | 10(17.5) | |
| COVID-19 | 0(0.00) | 7(31.8) | 7(12.3)) | |
| *Others | 5(14.3) | 4(18.2) | 9(15.8) | |
| Total | 35(100.0) | 22(100.0) | 57(100.0) | |
| | | | | |
| Final causes of death | | | | 0.092[f] |
| Cardio-pulmonary arrest | 11(31.4) | 11(50.0) | 22(38.6) | |
| Respiratory failure | 7(20) | 6(27.3) | 13(22.8) | |
| Renal failure | 1(2.9) | 1(4.5) | 2(3.5) | |
| Liver Failure | 0(0.00) | 1(4.5) | 1(1.8) | |
| Multiple organ failure | 16(45.7) | 3(13.6) | 19(33.3) | |
| Total | 35(100.0) | 22(100.0) | 57(100.0) | |

*Others comprised Kaposi sarcoma, advanced cervical cancer, suicide, ingestion of unsafe herbal medication, parasuicide , and supraventricular tachy-cardia. Early pregnancy bleeding includes those from ectopic pregnancy, miscarriages, and termination of pregnancy. Abbreviations: c, Chi-square test; f, Fisher exact test.

 

**Table 9. Patient, healthcare professional, and administrative-related avoidable factors.**

| Avoidable factor (AVF) | Value: n (%) | | | *p*-value |
|---|---|---|---|---|
| | Before COVID-19 | During COVID-19 | Total | |
| Presence of any AVF, n (%) | | | | |
| Present | 35(100.0) | 22(100.0) | 57(100.0) | |
| Absent | 0(0.0) | 0(0.0) | 0(0.0) | |
| Total | 35(100) | 22(100) | 57(100.0) | |
| | | | | |
| Patient-related AVF, n (%) | | | | 1.000 |
| Present | 22(62.9) | 8(36.4) | 30(52.6) | |
| Absent | 13(37.1) | 14(63.6) | 27(47.4) | |
| Total | 35(100.0) | 22(100.0) | 57(100.0) | |
| | | | | |
| Types of patient-related avoidable factors, n | | | | |
| Had no antenatal care | 13 | 3 | 16 | |
| Defaulted antenatal clinic | 0 | 1 | 1 | |
| Refused medical treatment | 0 | 0 | 0 | |
| Commenced antenatal clinic after 20 weeks' gestation | 6 | 1 | 7 | |
| Declined medical treatment | 2 | 2 | 4 | |
| Delayed seeking medical treatment | 11 | 4 | 15 | |
| Unsupervised delivery at home | 1 | 1 | 2 | |
| Total | 33 | 12 | 45 | |
| | | | | |
| Healthcare professional related AVF, n (%) | | | | 0.272 |
| Present | 22(62.9) | 12(54.5) | 34(59.6) | |
| Absent | 13(37.1) | 10(45.5) | 23(40.4) | |
| Total | 35(100.0) | 22(100.0) | 57(100.0) | |
| | | | | |
| Type of healthcare professional-related AVF, n | | | | |
| Lack of critical care skills | 7 | 4 | 11 | |
| Delay in referral | 7 | 2 | 9 | |
| Wrong diagnosis despite classical features of a disease | 6 | 1 | 7 | |
| Wrong treatment | 10 | 2 | 12 | |
| Delay in seeking senior outputs | 4 | 3 | 7 | |
| Total | 34 | 12 | 46 | |
| | | | | |
| Administrative-related AVF, n (%) | | | | 0.779 |
| Present | 20(57.1) | 11(50.0) | 31(54.4) | |
| Absent | 15(42.9) | 11(50.0) | 26(45.6) | |
| Total | 35(100.0) | 22(100.0) | 57(100.0) | |
| | | | | |
| Type of administrative-related AVF, n | | | | |
| Delay in referral (transportation) | 7 | 1 | 8 | |
| Communication breakdown | 5 | 3 | 8 | |
| Lack of equipment | 2 | 0 | 2 | |

*(Continued)*

**Table 9.** (Continued)

| Avoidable factor (AVF) | Value: n (%) | | | p-value |
|---|---|---|---|---|
| | Before COVID-19 | During COVID-19 | Total | |
| Lack of trained medical staff | 4 | 0 | 4 | |
| Lack of ICU and high care dependent unit space | 11 | 7 | 18 | |
| Lack of medicine | 0 | 1 | 1 | |
| Total | 29 | 12 | 41 | |

Multiple avoidable factors were associated with the management of some patients, and this made the calculation of percentages of the type of avoidable factor less informative. Additionally, some avoidable factors may be grouped into more than one category; however, the current grouping in this table was based on specific circumstances associated with each case after a robust panel discussion. Abbreviations: c, Chi-square test.

during the COVID-19 pandemic. The most frequent AVF associated with the management of HDP was a delay in referral to a tertiary hospital: 10 before and four during the COVID-19 pandemic.

## 4. Discussion

### 4.1 Main findings

Most of the women before and during the COVID-19 pandemic were aged 25–34 years, single, lived in the Kenneth Kaunda district, were unemployed, and neither smoked nor used any illicit substance. The number of maternal deaths and iMMRs was more before than during the COVID-19 pandemic. Both the maternal deaths and iMMR decreased yearly during the study period, with a 9.59% declining trend in the six-monthly iMMR between the first and last six months of the study period. The first, second, and third most common causes of maternal deaths before the COVID-19 pandemic were preeclampsia with severe features, tuberculosis, and early pregnancy bleeding, respectively. During the COVID-19 pandemic, the first and second most common causes of maternal death were COVID-19 and tuberculosis, respectively, with the former attributable to 31.8% (7/22) deaths. The third most common causes of maternal death during the COVID-19 pandemic were PPH, pulmonary embolism, sepsis, and preeclampsia with severe features. The percentage of the women who received mechanical ventilation was significantly higher before COVID-19 pandemic than that during the COVID-19 pandemic. Most maternal deaths occurred at night, with most hospital admissions occurring during the day.

Avoidable factors were associated with all the maternal deaths. Before the COVID-19 pandemic, the most frequent patient-, healthcare professional-, and administrative-related AVFs were failure to book for antenatal care, administration of wrong treatment, and lack of ICU/high care bed spaces, respectively. During the COVID-19 pandemic, the most common patient-, healthcare professional- and administrative-related AVFs were delay in seeking medical treatment, lack of critical care skills, and unavailability of ICU/high care bed spaces, respectively.

### 4.2 Interpretation

Most of the women who died were between the ages of 25 and 34. The predominant age group in this study was similar to the findings of Thomas et al. in which participants were between the ages of 14 and 45 [17]. In contrast, Rulisa et al. found that the most common age group was 16–57 years [18]. In non-pregnant adults, however, death due to COVID-19 affected mainly individuals aged 60 years and above [19].

In South Africa, some patients access healthcare across health districts due to reasons such as referral patterns, logistical concerns, and financial cost considerations [20]. In this study setting, the restriction of movements during the COVID-19 period did not affect the total deliveries and births because there was no decline in their rates (Tables 3 and 4). However, Bailey et al 2023 reported an increase in birth rates among native Americans during the COVID-19 pandemic [21]. In contrast,

**Table 10. Avoidable factors associated with the management of causes of maternal death.**

| Avoidable factors | Value: n | | |
|---|---|---|---|
| | Before COVID-19 | During COVID-19 | Total |
| Avoidable factors related to non-pregnancy-related infection, n | 14 | 10 | 24 |
| Defaulted TB treatment multiple times | 8 | 3 | 11 |
| CD4 not checked by a healthcare professional | 6 | 5 | 11 |
| Poor compliance with ARV treatment | 12 | 8 | 20 |
| | | | |
| Avoidable factors related to hypertensive disorders of pregnancy | 13 | 6 | 19 |
| Delay in referral to tertiary hospital | 10 | 4 | 14 |
| Pre-eclampsia screening test not performed healthcare professional | 9 | 2 | 11 |
| | | | |
| Avoidable factors related to obstetric haemorrhages | 5 | 3 | 8 |
| Problem recognition of bleeding ectopic pregnancy | 3 | 0 | 3 |
| INR not requested in IUFD | 1 | 0 | 1 |
| Poor surgical skills at caesarean delivery | 1 | 1 | 2 |
| Delay in taking a relook decision | 0 | 2 | 2 |
| | | | |
| Avoidable factors related to puerperal sepsis | 4 | 1 | 5 |
| Problem recognition of puerperal sepsis | 1 | 0 | 1 |
| Delay in deciding to perform TAH | 1 | 0 | 1 |
| Problem recognition of RPOC | 1 | 0 | 1 |
| | | | |
| Avoidable factors related to ectopic pregnancy. | 2 | 1 | 3 |
| Obvious features of ectopic pregnancy misdiagnosed and the patient received misoprostol for termination of pregnancy on request until hypovolemic shock occurred | 0 | 1 | 1 |
| Classical features of ectopic pregnancy missed at the medical ED where the patient was brought for symptomatic anemia, but urine beta hCG test was not performed | 1 | 0 | 1 |
| Classical features of ectopic pregnancy missed and treated as pelvic inflammatory disease in pregnancy | 1 | 0 | 1 |
| | | | |
| Avoidable factors related to medical and surgical conditions | 9 | 4 | 12 |
| Suicide | 0 | 1 | 1 |
| Late presentation of advanced cervical cancer in pregnancy | 2 | 1 | 3 |
| Failure to refer cardiac patients to a higher level of care | 2 | 2 | 4 |
| Inadequate critical care | 4 | 0 | 4 |
| Delay in diagnosis of obvious pelvic source of sepsis in the medical ward | 1 | 0 | 1 |

Multiple avoidable factors were associated with the management of some patients. Additionally, some avoidable factors may be grouped into more than one category; however, the current grouping in this table was based on specific circumstances associated with each case after a robust panel discussion. Abbreviations: ARV , Anti-retroviral drugs; ED , Emergency Department; INR , International normalized ratio; IUFD , Intrauterine fetal death; RPOC, Retained product of conception; TAH , Total abdominal hysterectomy; TB , Tuberculosis.

Wachuli et al. found that COVID-19 severely impacted maternity services in their setting and was associated with a reduction in the number of antenatal care revisits [22]. However, the index study setting was a referral centre that prioritized maternity services. Since most of the deceased were from the same district (Kenneth Kaunda) as the study setting, restrictions in movement/transportation might not have impacted the referral pattern before and during the COVID-19 pandemic.

Surprisingly, there was a decline in maternal deaths and iMMR during the COVID-19 pandemic as compared to the period before COVID-19. In South Africa, the overall iMMR per 100 000 live births was 98.8 in 2019 (before the COVID-19 pandemic), 126.1 in 2020, 148.1 in 2021, 109.9 in 2022, and 102.1 in 2023 [23]. Compared to the pre-pandemic levels, COVID-19 caused an increase in iMMR in 2020 and 2021 but there was a decline observed in 2022 [23]. Similarly, Beňová et al reported an increase in iMMR during the COVID-19 pandemic [24]. Most plausibly, the decline in maternal deaths and iMMR at the index study setting, despite the occurrence of the COVID-19 pandemic, might be because of continuous and sustained improvement in maternity care at the facility. The number of full-time specialist obstetricians and gynaecologists doubled from two to four in 2019 and was retained. Moreover, additional clinical staff were allocated to essential services such as maternity care during the COVID-19 pandemic. Nonetheless, the differences in the clinical profile of patients, coupled with the virulence of the SARS-COV-2 strain and other supportive care, might also have played a role in the differences in trends observed in various settings by different researchers. For instance, the rate of mechanical ventilation was lower during than before the COVID-19 pandemic. Given that COVID-19 is a highly infectious respiratory disease, the lower rate of mechanical ventilation during the pandemic may indicate that the deceased pregnant women who presented with COVID-19 in the study setting did not have severe respiratory complications requiring mechanical ventilation.

Notably, the first, second, and third most common causes of maternal deaths before the COVID-19 pandemic were preeclampsia with severe features, tuberculosis, and early pregnancy bleeding, respectively. This is similar to the findings of another study showing that among the most common causes of maternal deaths before the COVID-19 pandemic were sepsis, haemorrhage such as those from abortion, and HDP particularly preeclampsia [18]. During the COVID-19 pandemic, the first most common cause of maternal death was COVID-19, with tuberculosis being the second. Similar to our findings, the most common causes of maternal deaths during the COVID-19 pandemic have been reported by other investigators as COVID-19 pneumonia and associated acute respiratory distress [25].

All the maternal deaths were associated with an AVF, which could possibly have prevented some of the deaths. This information re-echoes the findings of the National Committee on Confidential Enquiry into Maternal Deaths in South Africa which showed that 58% of the maternal deaths in South Africa had been preventable [23]. During the COVID-19 pandemic, the most common patient, healthcare professional, and administrative-related AVFs in our study were delay in seeking medical treatment, lack of critical care skills, and lack of ICU/high care dependent unit bed spaces, respectively. Before the COVID-19 period, though, the most frequent patient, healthcare professional, and administrative-related AVFs were failure to book for antenatal care, administration of wrong treatment, and lack of ICU/high care dependent unit bed spaces, respectively. Notably, some of the AVFs were related to a specific clinical condition (Table 10), and this becomes a clarion call for development and adherence to clinical protocols. This need has been highlighted by Sadler et al. who showed that the management of obstetric conditions such as puerperal sepsis and PPH was associated with substandard care and lack of problem recognition [26].

## 4.3 Strengths and limitations

There were some missing data because of the retrospective design of the study. To address this limitation, missing data were reported and excluded from the calculation of p-values. The results of the histopathological autopsies were not available to the investigators, because in South Africa forensic autopsies are usually conducted for maternal deaths and the reports are only made available to the family members of the deceased. Moreover, there was a restriction in performing histopathological autopsy during the early period of the COVID-19 pandemic due to concerns about disease spread. Furthermore, data on any

maternal death that may have occurred outside the hospital setting such as at home were not available, and we are therefore unable to provide further comments about them. Additionally, the non-inclusion in the study of pregnant women who survived (including those with near misses) did not allow for comparison of the outcomes (diagnosis and interventions) in the deceased and survivors. However, our study design is acceptable, as maternal death analysis in many countries often does not include survivors [10, 23]. Furthermore, the sample size may impact our conclusions. However, some of the statistical tests utilized, such as Pearson's Chi-square distribution test and Fisher exact test, are robust for small sample sizes. To aid our conclusion, we conducted joinpoint regression modeling (also known as change point regression analysis) of the trend in maternal mortality ratio with the number of data points exceeding the required minimum of seven [16].

A major strength of our study is its comparative design, and the relatively long duration (four years) studied. This study is therefore among a few with these attributes, particularly among those conducted in low- and middle-income countries. Furthermore, the medical records of the maternal deaths were discussed by the investigators during scheduled meetings to ensure the appropriateness of the data collected. This approach prevents bias during data collection.

### 4.4 Recommendations and generalizability

The following are measures that may be implemented at the study setting and similar healthcare facilities to improve care. Firstly, staff training is required to prevent healthcare professional-related AVFs. Among the required training are critical care skills as well as diagnosis and treatment of common causes of maternal deaths. Clinical staff should be rostered to undergo practical simulated training in common obstetric conditions. An example is the Essential Steps in Managing Obstetric Emergencies training.

Secondly, there is a need to streamline administrative factors by improving the healthcare system. Notable among them is the need to provide additional bed spaces for women needing critical care. Thirdly, there is a need to retain staff to avoid losing the gains made (declining iMMR). This may be in the form of seamless human resource services to staff and improving other conditions of services that will facilitate staff growth. Fourthly, there is a need to investigate the preponderance of maternal deaths that occur at night to develop measures to address factors associated with them. It is possible that the nocturnal maternal deaths may be partly related to poor staffing at night because there are already afternoon and weekend specialist handover ward rounds at the study setting. Fifthly, it is crucial to educate pregnant women about the need for antenatal care. In South Africa, over 68.3% of pregnant women attend antenatal care before 20 weeks of pregnancy [27]; however, there is a need to aim at a 100% uptake. The use of community-based healthcare workers and preventing barriers to access to prenatal care will improve antenatal clinic attendance [5, 28].

## 5. Conclusion

The iMMR was lower during the COVID-19 pandemic than before pandemic, with a steady yearly decline. Improved resource availability particularly employment of specialist clinical staff before the COVID-19 pandemic and the preferential resource allocation to maternity services during the pandemic may be responsible for the decline in the iMMR. Therefore, a major determinant of the outcome of a clinical condition is the implementation of appropriate interventions.

## Acknowledgments

My sincere appreciation to the management at Klerksdorp/Tshepong Hospital, for granting permission to conduct this study.

## Author contributions

**Conceptualization:** Ongombe Lunda, Lawrence Chauke, Ghada Daef, Nnabuike Chibuoke Ngene.

**Data curation:** Ongombe Lunda, Lawrence Chauke, Nnabuike Chibuoke Ngene.

**Formal analysis:** Gbenga Olorunfemi, Nnabuike Chibuoke Ngene.

**Investigation:** Ongombe Lunda, Lawrence Chauke, Ghada Daef, Nnabuike Chibuoke Ngene.

**Methodology:** Ongombe Lunda, Lawrence Chauke, Ghada Daef, Gbenga Olorunfemi, Nnabuike Chibuoke Ngene.

**Project administration:** Ongombe Lunda.

**Supervision:** Lawrence Chauke, Ghada Daef, Nnabuike Chibuoke Ngene.

**Validation:** Ongombe Lunda, Lawrence Chauke, Gbenga Olorunfemi, Nnabuike Chibuoke Ngene.

**Visualization:** Ongombe Lunda.

**Writing – original draft:** Ongombe Lunda.

**Writing – review & editing:** Ongombe Lunda, Lawrence Chauke, Ghada Daef, Gbenga Olorunfemi, Nnabuike Chibuoke Ngene.

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
