## [Decision Letter · Decision Letter 0]

PONE-D-24-59651Maternal deaths before and during COVID-19 pandemic: Causes and avoidable factors in a tertiary hospital in South Africa, 2018‒2022PLOS ONE

Dear Dr. Lunda,

Thank you for submitting your manuscript to PLOS ONE. After careful consideration, we feel that it has merit but does not fully meet PLOS ONE’s publication criteria as it currently stands. Therefore, we invite you to submit a revised version of the manuscript that addresses the points raised during the review process.

We look forward to receiving your revised manuscript.

Kind regards,

Douglas Aninng Opoku, MPH

Academic Editor

PLOS ONE

**Journal Requirements:**

1. When submitting your revision, we need you to address these additional requirements. Please ensure that your manuscript meets PLOS ONE's style requirements, including those for file naming. The PLOS ONE style templates can be found at https://journals.plos.org/plosone/s/file?id=wjVg/PLOSOne_formatting_sample_main_body.pdf and https://journals.plos.org/plosone/s/file?id=ba62/PLOSOne_formatting_sample_title_authors_affiliations.pdf 2. We note that you have indicated that there are restrictions to data sharing for this study. For studies involving human research participant data or other sensitive data, we encourage authors to share de-identified or anonymized data. However, when data cannot be publicly shared for ethical reasons, we allow authors to make their data sets available upon request. For information on unacceptable data access restrictions, please see http://journals.plos.org/plosone/s/data-availability#loc-unacceptable-data-access-restrictions.  Before we proceed with your manuscript, please address the following prompts: a) If there are ethical or legal restrictions on sharing a de-identified data set, please explain them in detail (e.g., data contain potentially identifying or sensitive patient information, data are owned by a third-party organization, etc.) and who has imposed them (e.g., a Research Ethics Committee or Institutional Review Board, etc.). Please also provide contact information for a data access committee, ethics committee, or other institutional body to which data requests may be sent. b) If there are no restrictions, please upload the minimal anonymized data set necessary to replicate your study findings to a stable, public repository and provide us with the relevant URLs, DOIs, or accession numbers. Please see http://www.bmj.com/content/340/bmj.c181.long for guidelines on how to de-identify and prepare clinical data for publication. For a list of recommended repositories, please see https://journals.plos.org/plosone/s/recommended-repositories. You also have the option of uploading the data as Supporting Information files, but we would recommend depositing data directly to a data repository if possible. Please update your Data Availability statement in the submission form accordingly.

**Additional Editor Comments:**

1. Abstract should be formatted according to the journal guidelines

2. Study design should be a cross-sectional design utilizing …….

3. Line 116-117. The statement on data collection research purposes is very confusing. 1 June 2022 to 31 December 2022 is not up to 4 or two years

4. Line 120, “K/T” should be written in full

5. Inclusion and exclusion criteria should be clearly stated

6. Line 157 and 160, Fig should be written in full

Reviewers' comments:

Reviewer's Responses to Questions

**Comments to the Author**

1. Is the manuscript technically sound, and do the data support the conclusions?

Reviewer #1: Yes

Reviewer #2: Partly

Reviewer #3: Yes

2. Has the statistical analysis been performed appropriately and rigorously? 

Reviewer #1: I Don't Know

Reviewer #2: Yes

Reviewer #3: Yes

3. Have the authors made all data underlying the findings in their manuscript fully available?

Reviewer #1: Yes

Reviewer #2: Yes

Reviewer #3: Yes

4. Is the manuscript presented in an intelligible fashion and written in standard English?

Reviewer #1: Yes

Reviewer #2: Yes

Reviewer #3: Yes

5. Review Comments to the Author

**Reviewer #1:**  General Comments

Although the manuscript was written in standard English, the language should be improved.

Also, the date format used in the manuscript must be corrected (e.g. 11 March should become March 11).

Consistency: Different forms of words have been used in the text. E.g. 'post-partum' with a hyphen was used three times; 'postpartum' without a hyphen was used four times. Please pick one style and use it consistently throughout the text.

Specific Comments

Introduction:

Line 80 Recommend replacing "are still battling" with "still struggle"

Results:

line 196 Recommend changing "demise" to died

Part of Table 5 (line 292-294) looks clumsy. Must be made clear.

The first two columns of Table 7 (line 337-338) have to aligned.

Discussion:

Line 402 Recommend changing "in this order" to respectively.

Line 404 Recommend changing "third" to three.

Line 406 Recommend changing "had" to received.

Line 407 Recommend changing the phrase "ventilation was statistically more before than that during the COVID-19-P." to "ventilation was significantly higher before COVID-19-P than that during the COVID-19-P."

Line 420 Recommend changing "where" to "in which"

Line 421 Recommend changing "Contrarily" to "In contrast"

Line 428 Recommend changing "they were no decline" to "there was no decline"

Line 504 "professionrelated" must be corrected

Line 524 Recommend replacing "The iMMR was lower during than before the COVID-19 period" with "The iMMR was lower during the COVID-19 pandemic than before pandemic"

**Reviewer #2: ** The researchers should address issues of non-standardized abbreviations, e.g., "MDs" to mean maternal deaths and "COVID-19 P" to mean COVID-19 pandemic.

The total sample size of 57 maternal deaths is relatively small for a robust statistical comparison. This raises concerns about statistical power and the ability to detect meaningful differences between pre-pandemic and pandemic periods.

**Reviewer #3: ** The study sought to determine the trends in institutional maternal mortality rate (iMMR), causes, and associated avoidable factors (AVFs) associated with maternal deaths (MDs) in a tertiary hospital in the North West province in South Africa over a four-year period (1 March 2018–28 February 2022) comprising two years before and two years during the COVID-19

pandemic. The study design was robust and appropriate statistical analysis performed rigorously. The data obtained supports the conclusion. The authors highlighted the strengths and limitations of the study as well as some recommendations which seemed feasible.

Minor comments for Authors

Line 333: It should read 26/35 instead of 26/36

Line 364: It should read 'administrative-related' instead of 'patient-related'

6. PLOS authors have the option to publish the peer review history of their article (what does this mean? ). If published, this will include your full peer review and any attached files.

**Do you want your identity to be public for this peer review?** For information about this choice, including consent withdrawal, please see our Privacy Policy .

Reviewer #1: No

Reviewer #2: No

Reviewer #3: No

---

## [Author Response · Author response to Decision Letter 1]

23 Apr 2025

Professor Douglas Aninng Opoku

Academic Editor: PLOS ONE

Dear Editor,

Response to reviewers’ comments

Manuscript: Maternal deaths before and during COVID-19 pandemic: Causes and avoidable factors in a tertiary hospital in South Africa, 2018‒2022. [PONE-D-24-59651]

The authors are thankful for the consideration being given to our manuscript. We have read the comments by the reviewers, and have used them to improve the manuscript. This letter contains point-by-point responses to the reviewers’ comments. Changes made in the revised manuscript have been highlighted.

Journal Requirements:

Authors’ response: The PLOS ONE requirements have been met.

Changes made in the manuscript: The manuscript has been revised to meet PLOS ONE requirements.

2. We note that you have indicated that there are restrictions to data sharing for this study. For studies involving human research participant data or other sensitive data, we encourage

authors to share de-identified or anonymized data. However, when data cannot be publicly shared for ethical reasons, we allow authors to make their data sets available upon request.

For information on unacceptable data access restrictions, please see http://journals.plos.org/plosone/s/data-availability#loc-unacceptable-data-access-restrictions.

b) If there are no restrictions, please upload the minimal anonymized data set necessary to replicate your study findings to a stable, public repository and provide us with the relevant

URLs, DOIs, or accession numbers. Please see http://www.bmj.com/content/340/bmj.c181.long for guidelines on how to de-identify and prepare clinical data for publication. For a list of recommended repositories, please see https://journals.plos.org/plosone/s/recommended-repositories. You also have the option of uploading the data as Supporting Information files, but we would recommend depositing data directly to a data repository if possible.

Authors’ response: Data availability statement in the submission form has been updated accordingly.

Changes made in the manuscript: Data availability statement in the submission form has been updated. The current statements are:

The data cannot be shared publicly because it contains sensitive patients' information. The data underlying the results presented in the study are available via the following institutional contact after their permission:

University of the Witwatersrand Human Research Ethics Committee, Faculty of Health Sciences, Philip Tobias Building, Offices 301 – 304, 3rd Floor, Corner York Road and 29 Princess of Wales Terrace, Parktown, 2193, Gauteng Province, South Africa. Phone: +27 11 717 2700. Email: HREC-Medical.ResearchOffice@wits.ac.za. Access to the data does not involve a third-party organization.

Authors’ response: The references have been reviewed and they are correct. No retracted article was cited.

Changes made in the manuscript: None

Additional Editor Comments:

1. Abstract should be formatted according to the journal guidelines

Authors’ response: The recommended revision has been made.

Changes made in the manuscript: The abstract has been formatted according to the journal guidelines.

2. Study design should be a cross-sectional design utilizing …….

Authors’ response: Revision has been made.

Changes made in the manuscript: The statement about study design has been revised as: The study was a cross-sectional design utilizing retrospective medical record review of …

3. Line 116-117. The statement on data collection research purposes is very confusing. 1 June 2022 to 31 December 2022 is not up to 4 or two years

Authors’ response: The maternal deaths that occurred from March 1, 2018 to February 28, 2022 were studied using the medical records of the deceased women as source of data. These maternal deaths occurred over a 4-year period. Data extracted/collected from the medical records of these women commenced on June 1, 2022 and ended on December 31, 2022. The data collection therefore commenced after ethical approval was granted on March 25, 2022.

To explain further, following submission of the manuscript, PLOS ONE returned the manuscript and requested that the date of data collection should be included in the manuscript. Because of this requirement, the sentence has been revised to avoid confusion. However, the authors are willing to delete it if required.

Changes made in the manuscript: The following statement has been revised to avoid confusion: “Data collection from the medical records for research purposes was from 1 June 2022 to 31 December 2022.”

The revised version is “Data extracted/collection from the medical records of the deceased women commenced on June 1, 2022 and ended on December 31, 2022.”

4. Line 120, “K/T” should be written in full

Authors’ response: This suggested revision has been made.

Changes made in the manuscript: K/T has been written in full throughout the manuscript.

5. Inclusion and exclusion criteria should be clearly stated

Authors’ response: The inclusion and exclusion criteria have been stated under a new sub-heading “Inclusion and exclusion criteria.” However, these criteria were already included in Figure 1.

Changes made in the manuscript:

Under a new sub-heading “inclusion and exclusion criteria,” the following have been included:

The inclusion criteria were all pregnant women and those in the puerperium (within 42 days of childbirth) who died in the Klerksdorp/Tsphepong hospital complex. The exclusion criteria were: (i) maternal deaths that occurred outside Klerksdorp/Tshepong hospital complex. (ii) pregnant and postnatal women who died following accidental causes.

6. Line 157 and 160, Fig should be written in full

Authors’ response: This has been done

Changes made in the manuscript: Fig has been replaced with Figure throughout the manuscript.

Reviewers' comments:

Reviewer's Responses to Questions

Comments to the Author

1. Is the manuscript technically sound, and do the data support the conclusions?

The manuscript must describe a technically sound piece of scientific research with data that supports the conclusions. Experiments must have been conducted rigorously, with

appropriate controls, replication, and sample sizes. The conclusions must be drawn appropriately based on the data presented.

Reviewer #1: Yes

Reviewer #2: Partly

Reviewer #3: Yes

Authors’ response: We are thankful to the authors. The reviewers’ comments to the authors have been used to improve the manuscript.

Changes made in the manuscript: The reviewers’ comments to the authors have been used to improve the manuscript as described below in the section on “Review Comments to the Author.”

2. Has the statistical analysis been performed appropriately and rigorously?

Reviewer #1: I Don't Know

Reviewer #2: Yes

Reviewer #3: Yes

Authors’ response: We are thankful to the authors.

Changes made in the manuscript: None

3. Have the authors made all data underlying the findings in their manuscript fully available?

Reviewer #1: Yes

Reviewer #2: Yes

Reviewer #3: Yes

Authors’ response: The data cannot be publicly shared for ethical reasons; however, the authors will make the data sets available to investigators upon request.

Changes made in the manuscript: Data availability statement in the submission form has been updated accordingly.

4. Is the manuscript presented in an intelligible fashion and written in standard English?

Reviewer #1: Yes

Reviewer #2: Yes

Reviewer #3: Yes

Authors’ response: We are thankful to the authors for the commendation

Changes made in the manuscript: None

5. Review Comments to the Author

Reviewer #1:

General Comments

Although the manuscript was written in standard English, the language should be improved. Also, the date format used in the manuscript must be corrected (e.g. 11 March should become March 11).

Authors’ response: The date format has been corrected.

Changes made in the manuscript: The date format has been corrected throughout the manuscript.

Consistency: Different forms of words have been used in the text. E.g. 'post-partum' with a hyphen was used three times; 'postpartum' without a hyphen was used four times. Please pick one style and use it consistently throughout the text.

Authors’ response: The recommended revision has been made.

Changes made in the manuscript: The word post-partum has been changed to postpartum throughout the manuscript.

Specific Comments

Introduction:

Line 80 Recommend replacing "are still battling" with "still struggle"

Authors’ response: The recommended revision has been made.

Changes made in the manuscript: "are still battling" has been replaced with "still struggle”

Results:

line 196 Recommend changing "demise" to died

Part of Table 5 (line 292-294) looks clumsy. Must be made clear.

The first two columns of Table 7 (line 337-338) have to aligned.

Authors’ response: The recommended revisions have been made.

Changes made in the manuscript: Demise has been replaced with died. Table 5 has been made clear. The first two columns of Table 7 are now well-aligned.

Discussion:

Line 402 Recommend changing "in this order" to respectively.

Line 404 Recommend changing "third" to three.

Line 406 Recommend changing "had" to received.

Line 407 Recommend changing the phrase "ventilation was statistically more before than that during the COVID-19-P." to "ventilation was significantly higher before COVID-19-P than that during the COVID-19-P."

Line 420 Recommend changing "where" to "in which"

Line 421 Recommend changing "Contrarily" to "In contrast"

Line 428 Recommend changing "they were no decline" to "there was no decline"

Line 504 "professionrelated" must be corrected

Line 524 Recommend replacing "The iMMR was lower during than before the COVID-19 period" with "The iMMR was lower during the COVID-19 pandemic than before pandemic"

Authors’ response: Thank you. In the initial manuscript, we could not find the word "professionrelated." However, all other recommended changes have been made.

Changes made in the manuscript: The recommended changes have been made.

Reviewer #2:

The researchers should address issues of non-standardized abbreviations, e.g., "MDs" to mean maternal deaths and "COVID-19 P" to mean COVID-19 pandemic.

Authors’ response: In the manuscript, the abbreviations MD, and COVID-19-P have been replaced with maternal death and COVID-19 pandemic respectively.

Changes made in the manuscript: In the manuscript, the abbreviations MD, and COVID-19-P have been replaced with maternal death and COVID-19 pandemic respectively.

The total sample size of 57 maternal deaths is relatively small for a robust statistical comparison. This raises concerns about statistical power and the ability to detect meaningful differences between pre-pandemic and pandemic periods.

Authors’ response: We have stated in the limitations of the study that the sample size may impact our conclusions. However, some of the statistical tests utilized, such as Pearson’s Chi-square distribution test and Fisher exact test, are robust for small sample sizes. To aid our conclusion, we conducted joinpoint regression modeling (also known as change point regression analysis) of the trend in maternal mortality ratio with the number of data points exceeding the required minimum of seven.

Changes made in the manuscript: Under the sub-heading Strengths and limitations, the following has been added as the last sentence.

Furthermore, the sample size may impact our conclusions. However, some of the statistical tests utilized, such as Pearson’s Chi-square distribution test and Fisher exact test, are robust for small sample sizes. To aid our conclusion, we conducted joinpoint regression modeling (also known as change point regression analysis) of the trend in maternal mortality ratio with the number of data points exceeding the required minimum of seven.

Reviewer #3:

The study sought to determine the trends in institutional maternal mortality rate (iMMR), causes, and associated avoidable factors (AVFs) associated with maternal

deaths (MDs) in a tertiary hospital in the North West province in South Africa over a four-year period (1 March 2018–28 February 2022) comprising two years before and two years

during the COVID-19 pandemic. The study design was robust and appropriate statistical analysis performed rigorously. The data obtained supports the conclusion. The authors highlighted the strengths and limitations of the study as well as some recommendations which seemed feasible.

Authors’ response: Thank you much for the commendation.

Changes made in the manuscript: None

Minor comments for Authors

Line 333: It should read 26/35 instead of 26/36

Line 364: It should read 'administrative-related' instead of 'patient-related'

Authors’ response: Thank you very much. The corrections have been made.

Changes made in the manuscript:

(a) In the results, under the sub-heading “Time of hospital admission and hour of maternal death” we have replaced 26/36 with 26/35.

(b) In the results, under the sub-heading “Avoidable factors” we have replaced patient-related with administrative-related.

Option to publish the peer review history

6. PLOS authors have the option to publish the peer review history of their article (what does this mean?). If published, this will include your full peer review and any attached files. If you choose “no”, your identity will

---

## [Decision Letter · Decision Letter 1]

Maternal deaths before and during COVID-19 pandemic: Causes and avoidable factors in a tertiary hospital in South Africa, 2018‒2022

PONE-D-24-59651R1

Dear Dr. Lunda,

We’re pleased to inform you that your manuscript has been judged scientifically suitable for publication and will be formally accepted for publication once it meets all outstanding technical requirements.

Kind regards,

Douglas Aninng Opoku, MPH

Academic Editor

PLOS ONE

Additional Editor Comments (optional):

Reviewers' comments:

Reviewer's Responses to Questions

**Comments to the Author**

1. If the authors have adequately addressed your comments raised in a previous round of review and you feel that this manuscript is now acceptable for publication, you may indicate that here to bypass the “Comments to the Author” section, enter your conflict of interest statement in the “Confidential to Editor” section, and submit your "Accept" recommendation.

Reviewer #1: All comments have been addressed

Reviewer #3: All comments have been addressed

2. Is the manuscript technically sound, and do the data support the conclusions?

Reviewer #1: Yes

Reviewer #3: Yes

3. Has the statistical analysis been performed appropriately and rigorously? 

Reviewer #1: Yes

Reviewer #3: Yes

4. Have the authors made all data underlying the findings in their manuscript fully available?

Reviewer #1: Yes

Reviewer #3: Yes

5. Is the manuscript presented in an intelligible fashion and written in standard English?

Reviewer #1: Yes

Reviewer #3: Yes

6. Review Comments to the Author

Reviewer #1: (No Response)

Reviewer #3: (No Response)

7. PLOS authors have the option to publish the peer review history of their article (what does this mean? ). If published, this will include your full peer review and any attached files.

**Do you want your identity to be public for this peer review?** For information about this choice, including consent withdrawal, please see our Privacy Policy .

Reviewer #1: No

Reviewer #3: No

---

## [Editor Report · Acceptance letter]

PONE-D-24-59651R1

PLOS ONE

Dear Dr. Lunda,

I'm pleased to inform you that your manuscript has been deemed suitable for publication in PLOS ONE. Congratulations! Your manuscript is now being handed over to our production team.

Kind regards,

on behalf of

Dr. Douglas Aninng Opoku

Academic Editor

PLOS ONE